# Safeguarding children and strengthening families through the application and utilization of family group decision making: A systematic review protocol

Collen Mafira Ngadhi[1]*, Shenaaz Wareley[1], Patrick Joseph[1], Chanté Johannes[1], Gary Spolander[2], Nicolette Vanessa Roman[1]

1 University of the Western Cape, Centre for Interdisciplinary Studies of Children, Families, and Society, Cape Town, South Africa, 2 School of Applied Social Studies, Robert Gordon University, Aberdeen, Scotland

* nmcollen@gmail.com

## Abstract

Children and families often face adversities that require whole-family interventions to promote well-being. Traditional child welfare models often demonstrate insufficient family involvement in decision-making, leading to gaps in safeguarding efforts. The Family Group Decision-Making (FGDM) model emerged as an intervention that empowers families to identify and develop solutions to their challenges. The aim of this systematic review protocol is to assess the application and utilisation of FGDM with a specific focus on synthesizing evidence that can inform its implementation within developing country contexts, where child welfare systems may face unique challenges. The Population, Intervention, Comparison, Outcome (PICO) framework will be used to define the research question. The inclusion and exclusion criteria will be based on studies that focus on FGDM (including its variants), publications from 2014 to December 2024, and studies published in English. Qualitative and quantitative design studies will be included. EBSCOhost (CINAHL, Masterfile Premier, MEDLINE, Academic Search Ultimate), PubMed, Sabinet, Wiley Online Library, Scopus, and Web of Science will be accessed through the University of the Western Cape's library catalogue. Search terms will be combined with relevant keywords to create search strings using Boolean operators. The Critical Appraisal Skills Programme (CASP) checklists will be used for appraisal. Three independent reviewers will screen articles, a fourth reviewer consulted in case of disagreements. The Preferred Reporting Items for Systematic Reviews and Meta-Analysis (PRISMA) flowchart will illustrate the search, screening, and selection process. Data will be qualitatively synthesized using thematic analysis. This protocol is registered on Open Science Framework https://doi.org/10.17605/OSF.IO/B3TH8.

**Data availability statement:** No datasets were generated or analysed during the current study. All relevant data from this study will be made available upon study completion.

**Funding:** The author(s) received no specific funding for this work.

**Competing interests:** I have read the journal's policy and the authors of this manuscript have the following competing interests: The authors have declared that no competing interests exist.

## Introduction

Children and families often face challenges that require intervention to promote well-being [1]. The adversities require innovative intervention strategies, including a whole-family approach, strong practitioner-parent relationships, and comprehensive support services [2]. Yobouet [3] posits that the adversities encountered by families require a comprehensive strategy that incorporates the collaboration among governments, non-governmental organizations, and family units. Tyack [4] proposed that co-designing interventions is a more effective approach to addressing the challenges faced by children and families compared to using pre-designed interventions. Family Group Decision-Making (FGDM) has emerged as an effective approach in child welfare addressing child abuse and neglect involving extended family systems during critical decision-making process [5].

The FGDM approach has shifted from a problem-focused approach to a strength-based shared decision-making between clients and service providers [6]. The FGDM is a participatory process that empowers families to collaborate in decisions regarding the care and protection of their children [7], enhancing family cohesion and preventing child maltreatment [8]. Family Group Decision Making (FGDM), or Family Group Conferencing (FGC), is a collaborative approach that empowers families to participate in decision-making about child welfare. Ramon [9] posits that the approach is based on the belief that families can effectively address their challenges, leading to better outcomes for children and adults. Edwards, Parkinson [10] assert that FGCs are a type of family-led decision-making process in social care, emphasizing family expertise and participation. Hillebregt, Scholten [11] defined the FGDM as a systematic decision-making process designed to redistribute power from professionals and individuals in need and their families. Family Group Conferencing is a child welfare approach that brings together immediate and extended family members, along with others from the child's social network, to collaboratively determine the best strategy for addressing the needs of a child requiring support or protection [12].

Despite the implementation of FGDM, there is still a lack of understanding regarding its application, [6] including its use in safeguarding children and strengthening families. While Family Group Decision Making/Conferencing (FGDM/FGC) has been adopted in several high-income nations, for instance, New Zealand [13], the United States [7], the United Kingdom [10], and the Netherlands [14], empirical evidence from developing countries, particularly within Sub-Saharan Africa and other low- and middle-income context, remains scarce. The FGC model has been adopted in some of the high-income countries including Australia, Canada, Sweden, Norway and Denmark [15]. The majority of research originates from developed nations with well-resourced and institutionalized child welfare systems. Consequently, there is a limited synthesis of evidence specifically examining its application in developing countries. This results in a contextual gap in understanding how the model functions within resource-constrained child protection environments.

Existing scholarship and implementations, in South Africa, frequently concentrate on legislative frameworks, (the Children's Act 38 of 2005 and the Child Justice Act 75 of 2008) or restorative/child justice paradigms, rather than the broad application

of FGDM/FGC in routine child protection and family strengthening initiatives designed to safeguard against abuse and neglect. Moreover, developing nations in Sub-Saharan Africa, Southeast Asia, and Latin America present distinct family structures, community-based support systems and child protection challenges that may require modified approaches to FGDM. This geographical and contextual disparity underscores the need for a systematic synthesis of existing evidence from developing nations to inform culturally relevant and family-centered child welfare practices in resource-constrained environments. To address this limitation, the review will employ a subgroup analysis of studies conducted in low- and middle-income countries (LMICs) as defined by World Bank. Subgroup analysis will enable the synthesis of available evidence from these specific contexts while also leveraging the broader and more established body of research from high income countries to inform the overall analysis and identify key implementation factors relevant to resource-constrained settings. A major finding is that the FGDM has been diversified to include care of the elderly [11] and mental health [9]. Limited systematic review research has focused on the application of FGDM in the context of safeguarding children and strengthening families.

This study contributes to the existing research in many significant ways. First, it addresses the scarcity of FGDM research in developing nations where most empirical studies and systematic synthesis originate from high-income Western contexts. Second, this systematic review provides a comprehensive synthesis of current knowledge on FGDM, offering insights for policymakers, practitioners, and researchers. This review will inform future implementation strategies and help develop best practices in child protection. Finally, the findings will identify areas needing further research, guiding future efforts in family-centred child protection. The terms "Family Group Decision Making (FGDM)" and "Family Group Conferencing (FGC)" will be used interchangeably throughout this study. The aim of the study is to present a protocol for a systematic review examining the application of FGDM in safeguarding children and strengthening families, within child welfare contexts.

The research question will be:

Does the application of FGDM enhance child safety and family strengthening outcomes within the child welfare systems?

## Materials and methods

This systematic review protocol has been registered with the Open Science Framework (OSF) to ensure alignment with the intended methodology. The registration DOI is https://doi.org/10.17605/OSF.IO/B3TH8. In terms of amendment plan, there are no protocol amendment to date, any future amendment will be dated, documented in the OSF record and reported in the final systematic review. This research will rely on previously published data, so there will be no public or participant involvement in the design or dissemination of findings. The Preferred Reporting Items for Systematic Reviews and Meta-Analyses Protocols (PRISMA-P) checklist will be used to report the findings of this study.

### Inclusion criteria

The systematic review will include peer-reviewed articles, full-text journal articles, and theses focused on Family Group Decision Making (FGDM), Family Group Conference, or Family Group Conferencing. Participants should encompass children, parents (including guardian, foster or adoptive), family members, extended families and significant others. Studies published from 2014 to December 2024 will be considered as FGDM is an emerging field. Publications must be in English for systematic reviews due to difficulties in translating and replicating the review. Given the predominance of English-language publications we believe this restriction is unlikely to significantly alter our conclusions. A preliminary scoping search conducted in EBSCOHost, PubMed and Scopus revealed that approximately 85% of FGDM studies published between 2014 and 2024 are in English hence, this does not effect to the methodologies and outcome of the review's objectives. There will be no geographical restrictions applied to the selection of materials. To address the specific aim of understanding FGDM/FGC in developing country contexts, a subgroup analysis will be conducted on studies originating

from low- and middle-income countries as classified according to World Bank criteria. Both qualitative and quantitative studies will be included: qualitative studies for detailed insights, and quantitative studies for analyses of FGDM's effectiveness as an intervention.

## Exclusion criteria

Studies will be excluded if they are published in languages other than English, do not address topics related to FGDM, are repetitive or duplicates (appearing more than once), or involve participants outside the specified groups, such as those not including children, parents, families, extended families, or significant others.

## Search strategy

The following databases will be consulted, along with the rationale for selecting these databases: Web of Science, EBSCOhost: (CINAHL, Masterfile Premier, MEDLINE, Academic Search Ultimate), Sabinet, PubMed, Scopus, Wiley Online Library. Grey literature will be sourced through reference checking and contacting experts in the field of Family Group Decision Making, children safeguarding and family strengthening. The search process will involve a search string: ("Family Group Decision Making" OR "Family Group Conferencing" OR "Family Group Conference") AND ("Children" OR "Child Protection" OR "Families" OR "Family Strengthening" OR "Safeguarding"). The full search string will pilot tested to each database and filters include study types, dates and languages.

## Search terms

The search terms will include: Family Group Decision Making, Family Group Conferencing" Family Group Conference, children, child protection, families, family strengthening safeguarding. These search terms will be combined using Boolean operators for the selected databases. The search process runs from 2014 to December 2024. Additional sources, including the reference lists of included studies, will also be searched.

## Steps in the review process

**Identification of potential titles.**  The identification of potential titles will be done through deconstructing the research questions into fundamental concepts or search terms. A compilation of search terms pertaining to the key concepts will be established. The developed search terms or keywords will be conjoined with other relevant words to devise search strings utilizing the Boolean operators "AND", and "OR". ("Family Group Decision Making" OR "Family Group Conferencing" OR "Family Group Conference") AND (Children OR "Child Protection" OR Families OR "Family Strengthening" OR Safeguarding).

**Screening of studies.**  The screening of studies will be based on the inclusion and exclusion criteria outlined in this study. First, search results will be imported into Covidence, where duplicates will be removed. Next, title and abstract screening will exclude studies that do not meet the inclusion criteria. Studies that will have meet the inclusion criteria will undergo full-text review. The screening process will be conducted by three independent reviewers (CMN, SW and PJ) with a fourth reviewer (CJ) resolving conflicts and a fifth and sixth reviewer (GS and NVR) providing supervision.

**Full-text assessment.**  The methodological quality of the studies that meet the inclusion criteria will be assessed for inclusion within the systematic review. The Critical Appraisal Skills Programme (CASP) will function as a critical appraisal tool and is widely recognized as one of the most utilized quality assessment instruments, comprising ten questions designed to identify methodological deficiencies [16]. The three reviewers (CMN, SW and PJ) will independently assess each selected article and any disagreements regarding data extraction between the three reviewers will be resolved by a fourth reviewer (CJ) while the fifth (GS) and sixth reviewer (NVR) provides supervision. When disagreements arise, the involvement of a third independent reviewer is essential for resolution [17].

**Data extraction.** An Excel data extraction sheet will be designed and utilised to systematically extract data from all studies included in the review. The study will adopt the data extraction tool utilized by [18]. Data that is available from the extraction tool includes: study details (author(s), year of publication, study title, study purpose), participants details (sample size, demographics), methods (design and setting), data collection measures, main findings, implications and recommendations. Disagreements between the three reviewers regarding data extraction will be resolved by a fourth reviewer (CJ).

Preferred Reporting Items for Systematic Reviews and Meta-Analysis (PRISMA) flowchart detailing the search, screening and selection process will be employed to visually present the screening steps in the review process. It is a flow diagram that illustrates the steps involved in systematic review, including identification, screening, evaluation of eligibility, inclusion and exclusion [17]. The Preferred Reporting Items for Systematic Reviews and Meta-Analyses (PRISMA) flowchart will be employed to illustrate the identification, screening, eligibility, and inclusion phases of the systematic review process (S1 Fig).The outcomes of the study are mainly focused on child safeguarding, family strengthening and family cohesion.

The outcomes are defined as follows;

**Child Safeguarding:** Encompasses measures and interventions aimed at protecting children from maltreatment, including abuse, neglect, exploitation, and violence, ensuring their safety and well-being.

**Family Strengthening:** Involves enhancing the capacity of families to function effectively as a unit, promoting resilience, and supporting the family system to address challenges collaboratively.

**Family Cohesion:** Refers to the strengthening of relationships and bonds within the family unit, including immediate and extended family members, to foster unity and collaborative decision-making.

**Documenting the search process.** An Excel sheet will be created to keep track of all the activities regarding the systematic review process. The details will include databases consulted, search terms used, number of hits and the number of retrievals. Additionally, the appraisal tool, appraised articles and the data extraction sheet will be included on the excel sheet. All results that align with the specified outcome domains (child safeguarding, family strengthening, family cohesion, preventive approach) will be systematically extracted, encompassing all relevant measures, time points, and analyses detailed in the included studies. In instances where multiple measures or time points are reported, the primary outcomes as delineated by the study authors will be prioritized. Should there be ambiguity regarding the primary outcomes, the reviewers will select results based on their pertinence to the PICO framework, with all decisions meticulously documented and justified in the data extraction sheet.

## Data synthesis

The selected studies will be qualitatively synthesized, using thematic analysis to present their outcomes in the final review. Thematic analysis process involves identify common themes across studies and it involves coding the data and grouping codes into broader themes [19]. Thematic analysis seeks to identify themes within a dataset, thereby offering a comprehensive overview of the data and facilitating the emergence of new meanings and interpretations from the findings. The study will not undertake formal quantitative synthesis or meta-analysis hence, no quantitative effect measure will be calculated for the outcome of children safeguarding. Hence it will provide the descriptive numerical summaries, tabulate frequency distribution and thematic analysis. Qualitative studies and RCTs will be analyzed separately, as necessary, to account for methodological differences. The findings will be integrated into a narrative synthesis to provide a comprehensive overview of the application and outcomes of FGDM. The Cochrane Risk of Bias 2 tool will be implemented to help assessing certainty of evidence. For each synthesis to be eligible, we will attend to studies based on intervention characteristics (e.g., FGDM vs. FGC, implementation setting), study design (qualitative, quantitative, or mixed-methods), and outcome domains (child safeguarding, family strengthening, family cohesion, preventive approach). These characteristics will be compared to the planned syntheses (e.g.,

thematic synthesis for qualitative data on FGDM processes, narrative synthesis for quantitative outcomes like maltreatment reduction) using a summary table. Only studies relevant to each synthesis objective will be included, and decisions will be documented in the final review. To assess the robustness of synthesized results, sensitivity analyses will exclude studies rated as low quality using CASP checklists to determine their influence on thematic findings for child safeguarding and family strengthening.

## Discussion

The FGDM has its origins in New Zealand and has been used as a mechanism in child protection issues. Family Group Decision-Making (FGDM) is a commonly employed methodology in the domains of child welfare and social services. This approach emphasizes the active involvement of family members and their social networks in the decision-making processes pertaining to the well-being and welfare of children or other family members [20]. The FGDM model has been applied in child abuse, neglect and youth offending [5]. This approach has been employed in diverse settings, including child welfare, adult services, medical rehabilitation, youth offending, and mental health services [7]; elderly care, and the care of individuals with dementia or mental health [11]. The approach has also been applied in the care of mental health clients in public mental health care [9,21]. Based on the preceding analysis, it can be posited that the Family Group Decision-Making (FGDM) model has been employed in addressing child welfare issues; however, its application has also expanded to encompass various factors that impact families and necessitate their involvement in mitigating the adversities they face. The findings of the study are expected to contribute to the literature in the field of child protection and family strengthening. Furthermore, the findings are anticipated to benefit social services agencies that focus on children, child safeguarding, families and family strengthening providing best possible mechanism that can be applied in families and enhancing family involvement in welfare initiatives. The findings from this systematic review will assist policymakers in the developing family involvement intervention that prioritize the application and utilisation of FGDM in safeguarding children and strengthening families.

## Limitations

The predominance of FGDM research originating from developed countries may constrain the availability of studies pertaining to developing country, thereby affecting the granularity of subgroup analysis. This predominance as noted in preliminary scoping, may stem from implementation deficits and limitations in research capacity. The final review will document the geographic distribution of included studies, and address the potential implications of any underrepresentation of developing country contexts.

## Supporting information

**S1 Fig. PRISMA Flow Chart.**
(TIF)

## Author contributions

**Conceptualization:** Collen Mafira Ngadhi, Shenaaz Wareley, Chanté Johannes, Gary Spolander, Nicolette Vanessa Roman.

**Data curation:** Collen Mafira Ngadhi, Shenaaz Wareley, Patrick Joseph, Chanté Johannes, Gary Spolander, Nicolette Vanessa Roman.

**Formal analysis:** Collen Mafira Ngadhi, Shenaaz Wareley, Patrick Joseph, Chanté Johannes, Gary Spolander, Nicolette Vanessa Roman.

**Investigation:** Collen Mafira Ngadhi, Shenaaz Wareley, Patrick Joseph, Gary Spolander, Nicolette Vanessa Roman.

**Methodology:** Collen Mafira Ngadhi, Shenaaz Wareley, Patrick Joseph, Chanté Johannes, Gary Spolander, Nicolette Vanessa Roman.

**Project administration:** Collen Mafira Ngadhi, Gary Spolander.

**Resources:** Nicolette Vanessa Roman.

**Software:** Collen Mafira Ngadhi, Shenaaz Wareley, Patrick Joseph, Chanté Johannes, Gary Spolander, Nicolette Vanessa Roman.

**Supervision:** Gary Spolander, Nicolette Vanessa Roman.

**Validation:** Collen Mafira Ngadhi, Shenaaz Wareley, Patrick Joseph, Chanté Johannes, Gary Spolander, Nicolette Vanessa Roman.

**Visualization:** Collen Mafira Ngadhi, Patrick Joseph, Chanté Johannes.

**Writing – original draft:** Collen Mafira Ngadhi, Shenaaz Wareley, Patrick Joseph, Chanté Johannes, Gary Spolander, Nicolette Vanessa Roman.

**Writing – review & editing:** Collen Mafira Ngadhi, Shenaaz Wareley, Patrick Joseph, Chanté Johannes, Gary Spolander, Nicolette Vanessa Roman.

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
