## [Decision Letter · Decision Letter 0]

22 Jan 2026

Dear Dr. Ngadhi,

Thank you for submitting your manuscript to PLOS ONE. After careful consideration, we feel that it has merit but does not fully meet PLOS ONE’s publication criteria as it currently stands. Therefore, we invite you to submit a revised version of the manuscript that addresses the points raised during the review process.

https://journals.plos.org/plosone/s/submission-guidelines#loc-laboratory-protocols. Additionally, PLOS ONE offers an option for publishing peer-reviewed Lab Protocol articles, which describe protocols hosted on protocols.io. Read more information on sharing protocols at . Additionally, PLOS ONE offers an option for publishing peer-reviewed Lab Protocol articles, which describe protocols hosted on protocols.io. Read more information on sharing protocols at . Additionally, PLOS ONE offers an option for publishing peer-reviewed Lab Protocol articles, which describe protocols hosted on protocols.io. Read more information on sharing protocols at . Additionally, PLOS ONE offers an option for publishing peer-reviewed Lab Protocol articles, which describe protocols hosted on protocols.io. Read more information on sharing protocols at https://plos.org/protocols?utm_medium=editorial-email&utm_source=authorletters&utm_campaign=protocols....

We look forward to receiving your revised manuscript.

Kind regards,

Daryl Higgins, PhD

Academic Editor

PLOS One

**Journal Requirements:**

1. When submitting your revision, we need you to address these additional requirements. Please ensure that your manuscript meets PLOS ONE's style requirements, including those for file naming. The PLOS ONE style templates can be found at https://journals.plos.org/plosone/s/file?id=wjVg/PLOSOne_formatting_sample_main_body.pdf and https://journals.plos.org/plosone/s/file?id=ba62/PLOSOne_formatting_sample_title_authors_affiliations.pdf 2. Thank you for stating the following in your Competing Interests section:  “I have read the journal's policy and the authors of this manuscript have the following competing interests: The authors have declared that no competing interests exist”  Please complete your Competing Interests on the online submission form to state any Competing Interests. If you have no competing interests, please state "The authors have declared that no competing interests exist.", as detailed online in our guide for authors at http://journals.plos.org/plosone/s/submit-now This information should be included in your cover letter; we will change the online submission form on your behalf. 3. When completing the data availability statement of the submission form, you indicated that you will make your data available on acceptance. We strongly recommend all authors decide on a data sharing plan before acceptance, as the process can be lengthy and hold up publication timelines. Please note that, though access restrictions are acceptable now, your entire data will need to be made freely accessible if your manuscript is accepted for publication. This policy applies to all data except where public deposition would breach compliance with the protocol approved by your research ethics board. If you are unable to adhere to our open data policy, please kindly revise your statement to explain your reasoning and we will seek the editor's input on an exemption. Please be assured that, once you have provided your new statement, the assessment of your exemption will not hold up the peer review process. 4. If the reviewer comments include a recommendation to cite specific previously published works, please review and evaluate these publications to determine whether they are relevant and should be cited. There is no requirement to cite these works unless the editor has indicated otherwise. 

Reviewers' comments:

**Comments to the Author**

1. Does the manuscript provide a valid rationale for the proposed study, with clearly identified and justified research questions?

Reviewer #1: Yes

2. Is the protocol technically sound and planned in a manner that will lead to a meaningful outcome and allow testing the stated hypotheses?

Reviewer #1: Yes

3. Is the methodology feasible and described in sufficient detail to allow the work to be replicable?

Reviewer #1: Yes

4. Have the authors described where all data underlying the findings will be made available when the study is complete?

The PLOS Data policy requires authors to make all data underlying the findings described in their manuscript fully available without restriction, with rare exception, at the time of publication. The data should be provided as part of the manuscript or its supporting information, or deposited to a public repository. For example, in addition to summary statistics, the data points behind means, medians and variance measures should be available. If there are restrictions on publicly sharing data—e.g. participant privacy or use of data from a third party—those must be specified.requires authors to make all data underlying the findings described in their manuscript fully available without restriction, with rare exception, at the time of publication. The data should be provided as part of the manuscript or its supporting information, or deposited to a public repository. For example, in addition to summary statistics, the data points behind means, medians and variance measures should be available. If there are restrictions on publicly sharing data—e.g. participant privacy or use of data from a third party—those must be specified.

Reviewer #1: Yes

5. Is the manuscript presented in an intelligible fashion and written in standard English?

Reviewer #1: Yes

You may also provide optional suggestions and comments to authors that they might find helpful in planning their study.

**Reviewer #1:** There is a need to be specific concerning FGC focused on child care in developing countries .There is a need to be specific concerning FGC focused on child care in developing countries .There is a need to be specific concerning FGC focused on child care in developing countries .There is a need to be specific concerning FGC focused on child care in developing countries .

.

Reviewer #1: **Yes:**a.Prof. Shulamit Ramona.Prof. Shulamit Ramona.Prof. Shulamit Ramona.Prof. Shulamit Ramon

You may also use PLOS’s free figure tool, NAAS, to help you prepare publication quality figures: https://journals.plos.org/plosone/s/figures#loc-tools-for-figure-preparation

---

## [Author Response · Author response to Decision Letter 1]

18 Mar 2026

1. The manuscript has been formatted in accordance with PLOSONE guidelines, including the title page structure (title, authors and affiliations denoted via superscript numerals, corresponding author contact information) and the overall layout as specified in the provided template.

2. We confirm the statement “The authors have declared that no competing interest exist.” This has been updated in the online submission form and included in the manuscript.

3. In accordance with protocol, no primary data were generated for this review. Extracted data from the final review will be made fully and openly available deposited in the Open Science Framework (OSF) repository linked to our protocol registration. This commitment is explicitly stated in the Data availability statement included in the manuscript (end of manuscript, before references).

4. While the reviewer did not suggest specific publications, we incorporated additional relevant references to provide greater specificity regarding developing countries. We thoroughly reviewed the reference list, ensuring all citations are complete and relevant. No retracted papers have been cited.

5. The reference list was reviewed for completeness and accuracy. No retracted publications were cited. Minor revisions including the addition of digital object identifiers (DOI) where available and the standardization of Vancouver-style formatting. The modifications are detailed in the tracked changes version of the manuscript.

Reviewer comments:

Reviewer comment: “There is a need to be specific concerning FGC focused on child care in developing countries.”

The authors express their gratitude to Prof. Shulamit Ramon for her positive assessment of the protocol’s rationale, technical soundness, feasibility, replicability, data availability and presentation. The suggestion to enhance specificity regarding Family Group Conference (FGC) focused on child care in developing countries is gratefully acknowledged. To address this limitation, the manuscript has been revised in certain sections to emphasize on application of FGDM/FGC in child safeguarding and family strengthening within developing country contexts. For instance, the abstract and introduction highlights unique prevalent in developing nations (resource limitations, informal support mechanisms and implementation deficits).

However, we wish to address an important methodological consideration regarding the search strategy. While the reviewer raises a valid point concerning the focus on developing countries, we must balance this specificity against the potential risk of excluding relevant literature that could inform developing country contexts. Limiting the scope to developing country terminology may inadvertently exclude studies conducted in developing nations that offer discussions of implications for low-resource setting or cultural adaptations pertinent to non-Western context. The methods section (eligibility criteria and search strategy) did not incorporate this level of specificity as it may limit the number of eligible studies, potentially resulting in a smaller sample size for inclusion.

To reconcile the reviewer's emphasis on developing country specificity with the need to avoid excluding potentially relevant evidence, a two-pronged methodological approach will be adopted. Firstly, a broad primary search without geographic restrictions will capture the entirety of the extant FGDM literature. This approach is essential because restricting the search to developing country terms risks excluding relevant studies from developed nations, such as research on cultural adaptations with minority populations, implementation in resource-constrained community settings, and insights into family engagement that transcend national boundaries. Secondly, a layered analysis framework will address the developing country focus. Initially, all studies from low- and middle-income countries (as classified by the World Bank) will be flagged for primary subgroup analysis, thus providing direct evidence from developing contexts. Subsequently, a secondary analysis will examine developed country studies to identify findings transferable to developing settings, including explicit implications for low-resource environments and cultural adaptations relevant to non-Western family structures. Lastly, a comparative analysis will identify similarities and differences in FGDM outcomes across contexts.

---

## [Editor Report · Decision Letter 1]

24 Mar 2026

Safeguarding children and strengthening families through the application and utilization of family group decision making: A systematic review protocol.

PONE-D-25-52635R1

Dear Dr. Ngadhi,

We’re pleased to inform you that your manuscript has been judged scientifically suitable for publication and will be formally accepted for publication once it meets all outstanding technical requirements.

Kind regards,

Daryl Higgins, PhD

Academic Editor

PLOS One
---

## [Editor Report · Acceptance letter]

PONE-D-25-52635R1

PLOS One

Dear Dr. Ngadhi,

I'm pleased to inform you that your manuscript has been deemed suitable for publication in PLOS One. Congratulations! Your manuscript is now being handed over to our production team.

Kind regards,

on behalf of

Professor Daryl Higgins

Academic Editor

PLOS One